# Effects of Blanching, Freezing and Canning on the Carbohydrates in Sweet Corn

**DOI:** 10.3390/foods12213885

**Published:** 2023-10-24

**Authors:** Monica M. Whent, Holly D. Childs, Shawn Ehlers Cheang, Jiani Jiang, Devanand L. Luthria, Michael R. Bukowski, Carlito B. Lebrilla, Liangli Yu, Pamela R. Pehrsson, Xianli Wu

**Affiliations:** 1Methods and Application of Food Composition Laboratory, U.S. Department of Agriculture, Agricultural Research Service, Beltsville Human Nutrition Research Center, Beltsville, MD 20705, USA; monica.whent@usda.gov (M.M.W.); dave.luthria@usda.gov (D.L.L.); michael.bukowski@usda.gov (M.R.B.); pamela.pehrsson@usda.gov (P.R.P.); 2Department of Nutrition and Food Science, University of Maryland, College Park, MD 20742, USAlyu5@umd.edu (L.Y.); 3Department of Chemistry, University of California Davis, Davis, CA 95616, USA; ycheang@ucdavis.edu (S.E.C.); jsojiang@ucdavis.edu (J.J.); cblebrilla@ucdavis.edu (C.B.L.); 4Foods for Health Institute, University of California Davis, Davis, CA 95616, USA

**Keywords:** sweet corn, carbohydrate, dietary fiber, glycomic analysis, blanching, freezing, canning

## Abstract

Sweet corn is frequently consumed in the US and contains carbohydrates as major macronutrients. This study examined the effects of blanching, freezing, and canning on carbohydrates in sweet corn. Fresh bi-color sweet corn was picked in the field and processed immediately into frozen and canned samples. Simple sugars, starch, and dietary fiber (DF) (including total DF (TDF), insoluble DF (IDF) and two fractions of soluble DF (SDF)) were measured according to the AOAC methods. Additional glycomic analysis including oligosaccharides, monosaccharide composition of total polysaccharides (MCTP) and glycosidic linkage of total polysaccharides (GLTP) were analyzed using UHPLC-MS. Sucrose is the major simple sugar, and IDF is the main contributor to TDF. Sucrose and total simple sugar concentrations were not altered after blanching or freezing but were significantly reduced in canned samples. Kestose was the only oligosaccharide identified in sweet corn and decreased in all heat-treated or frozen samples. Starch content decreased in frozen samples but increased in canned samples. While two SDF fractions did not differ across all samples, blanching, freezing and canning resulted in increases in TDF and IDF. Six monosaccharides were identified as major building blocks of the total polysaccharides from MCTP analysis. Glucose and total monosaccharide concentrations increased in two canned samples. GLTP was also profoundly altered by different food processing methods. This study provided insights into the changes in the content and quality of carbohydrates in sweet corn after food processing. The data are important for accurate assessment of the carbohydrate intake from different sweet corn products.

## 1. Introduction

Sweet corn is a highly consumed vegetable in the United States [1]. It has also grown in popularity worldwide due to its pleasant taste and nutrient composition [2]. Carbohydrates are major macronutrients in sweet corn [3]. Roughly 20 g/100 g total carbohydrates were found in fresh sweet corn, containing mainly sugars, starch and dietary fiber [4]. Sweet corn is thus an important contributor of carbohydrates from the typical diet for Americans. Carbohydrates in foods are diverse molecules that range from simple sugars to highly complex polysaccharides [5]. According to the Dietary Guidelines for Americans, adults should consume between 45–65% of daily calories as carbohydrates, emphasizing plant foods such as whole grains and vegetables [6]. Carbohydrates are the largest component of energy in the human diet, accounting for 55% to 80% of total daily energy [7]. However, the role of plant-based carbohydrates goes beyond energy. Of the complex carbohydrates present in plant foods, only part of starch and some oligosaccharides are readily digestible within the human gastrointestinal (GI) tract. The non-digestible oligo- and polysaccharides, categorized as dietary fiber (DF), contribute to human health and GI function through other mechanisms. DF may act as a bulking agent to aid laxation, form gels to increase satiety and slow gastric emptying [8] and support commensal microbial communities in the lower GI tract, thereby impacting composition and host physiology [9]. Adequate DF intake may reduce the incidence of many chronic health conditions [10,11,12,13,14,15]. Nevertheless, it was estimated that only about 5% of Americans consume sufficient DF [16]. In addition, carbohydrates are major determining factors of the consumer acceptability of plant foods. The composition of carbohydrates in the endosperm affects the texture and mouthfeel of sweet corn [17].

Sweet corn is generally consumed in fresh or processed forms in the US. During the summer harvest, sweet corn can be purchased fresh and consumed after cooking. During the remaining seasons, sweet corn is mostly consumed in the canned or frozen form. It has long been known that food processing can alter the content and physicochemical properties of carbohydrates, resulting in changes in structural and physical properties that influence bio-accessibility and digestibility in the gastrointestinal (GI) tract, and further impact their physiological functions and health benefits [18,19,20]. For instance, almost 40 years ago Brand et al. demonstrated that minimally processed rice, potatoes and corn elicited a lower glycemic response in feeding processed foods such as instant rice, puffed rice, cornflakes, and instant potatoes [21]. Our group recently published a paper on the impacts of domestic cooking methods on carbohydrates in sweet corn [22]. We demonstrated that boiling and steaming did not alter the content of water-soluble carbohydrates such as simple sugars and sugar alcohols when compared to raw sweet corn, but resistant starch and total dietary fiber were reduced. While our findings were representative of different fresh sweet corn preparations, they offer a limited window into the carbohydrate composition of sweet corn which is also extensively consumed as a canned or frozen product.

The present study expands on our work, including the effects of blanching, freezing and canning on different carbohydrate fractions in sweet corn. The goal is to better understand the qualitative and quantitative changes of these components in sweet corn after food processing in a systematic way. The data from this study can be used for a more accurate assessment of the carbohydrate intakes from different sweet corn products, and to further establish the diet and health relationships impacted by food processing.

## 2. Materials and Methods

### 2.1. Plant Materials

Supersweet bicolor sweet corn (“Kickoff” variety) was harvested in September 2021 from Tuscarora Farm, MD. The corn was handpicked and selected based on color and size to ensure uniformity.

### 2.2. Chemicals and Reagents

Trifluoroacetic acid (TFA), chloroform (HPLC grade), ammonium acetate, ammonium hydroxide solution (NH_4_OH) (28–30%), sodium hydroxide (semiconductor grade, 99.99% trace metals basis), dichloromethane, anhydrous dimethyl sulfoxide (DMSO), iodomethane, 3-methyl-1-phenyl-2-pyrazoline-5-one (PMP), methanol (MeOH, HPLC grade), fructose, glucose, maltose, sucrose, and raffinose were purchased from Sigma-Aldrich (St. Louis, MO, USA). Maltotetraose, maltopentaose, maltohexaose, ketose, stachyose, and verbascose were obtained from Megazyme (Bray, Ireland). Acetonitrile (HPLC grade) was purchased from Honeywell (Muskegon, MI, USA). Sodium hydroxide (reagent grade), acetic acid (glacial), d-sorbitol, mannitol, deionized water, and ethanol (99%) were obtained from Fisher Scientific (Pittsburg, PA, USA).

### 2.3. Sweet Corn Processing Methods

Upon harvest, corn samples were immediately transported to the laboratory for processing. Freshly harvested corn was processed within a few hours into canned and two types of frozen samples as previously described [23]. Six samples were taken throughout the food processing workflow (Figure 1), including baseline sample (Baseline), blanched sample (Blanch), canned sample immediately after canning (Can 0), canned sample stored at ambient temperature for one month (Can 1), frozen sample with kernels on the cob stored for one month at −20 °C (FC 1) and frozen sample with kernels off the cob stored for one month at −20 °C (FO 1). Each processing method was performed in triplicate. For Baseline, Blanch, and FC 1 samples, kernels were manually taken off the cob using a corn stripper. For the two canned samples, the can contents were drained. The kernels of each sample were mixed with liquid nitrogen in a Vitamix blender (model VM0103, Cleveland, OH, USA) to homogenize the sample. The blended corn was freeze-dried immediately and stored until analysis at −20 °C (Figure 2).

### 2.4. Analyses by AOAC Methods

Measurements of different fractions of carbohydrates (sugar, starch and DF) in lyophilized sweet corn samples were performed at a certified commercial laboratory (Eurofins Food Integrity & Innovation, Madison, WI, USA) using standard methods of analysis (Figure 2). Simple sugars (mono-and disaccharides including fructose, glucose, sucrose, galactose, maltose, isomaltulose, and lactose) were assayed based on AOAC 2018.16 [24], and total simple sugar was calculated as the sum of all simple sugars. Briefly, sugars were extracted from sweet corn samples with a mixture of equal parts of water and ethanol, with the addition of an internal standard. Samples were filtered through a 10 kDa molecular weight filter, supernatants were dried and reconstituted for analysis by high-performance anion-exchange chromatography with pulsed amperometric detection (HPAEC-PAD). Starch was measured enzymatically according to AOAC 996.11 [25]. Freeze-dried sweet corn samples were extracted with ethanol to remove sugars, then the remaining residue was hydrolyzed into glucose with α-amylase and amyloglucosidase in pH 5 sodium acetate buffer solution. The resultant glucose was assayed enzymatically using the glucose oxidase/peroxidase (GOPOD) method. DF was determined by enzymatic-gravimetric-liquid chromatography approach (AOAC 2011.25) [26]. This method is applicable for the determination of different fractions of DF (insoluble DF (IDF), DF soluble in water but not in 78% alcohol (SDFP), DF soluble in water and 78% alcohol (SDFS)) and total DF (TDF, sum of IDF, SDFP and SDFS) inclusive of resistant starch [27].

To ensure data quality, the sweet corn samples were batched with quality control (QC) and certified reference materials for the analysis of each component. The QC materials had established tolerance limits used in USDA food composition studies [28]. A mixed flour QC material developed internally was analyzed for starch. Standard reference material (SRM) (SRM^®^ 3233 Breakfast Cereal) was procured from the National Institute of Standards and Technology (NIST) (Gaithersburg, MD) [29] and was used for quality control in the measurement of sugars and DF. Results for the SRM and flour QC material were evaluated relative to the expected tolerance limits for the flour QC material and the range from the certificate of analysis for SRM. Freeze-dried sweet corn samples were analyzed in two batches for simple sugars, starch and DF and the quality control materials were included and assayed with each batch of samples.

### 2.5. Sample Preparation for Glycomic Analysis

Lyophilized sweet corn materials were used for glycomic analysis adapted from published methods [30]. In short, 25 mg of lyophilized sweet corn powder was weighed to a 1.5 mL screw cap Eppendorf tube, and 1 mL of 80% ethanol was added to make a 25 mg/mL sample stock. Each sample was vortexed at the speed of 4 m/s for 2 min in a bead mill shaker and then centrifuged for 10 min at 10,000 rpm. After centrifugation, 900 µL solution was aliquoted from the supernatant and transferred to a clean 1.5 mL Eppendorf tube, and dried overnight in a vacuum concentrator for oligosaccharide analysis (Figure 2). The remaining pellet was rinsed with 1 mL 80% EtOH three times and then dried overnight in a vacuum concentrator to analyze the monosaccharide composition of total polysaccharides (MCTP) and glycosidic linkage of total polysaccharides (GLTP) (Figure 2).

### 2.6. Analysis of Oligosaccharides

The analytical method was adapted from a previous publication [30]. Briefly, the dried sample was reconstituted with 0.8 mL of water and diluted 400-fold in 75% acetonitrile. A 5 µL was then injected into an Agilent 1290 Infinity II UHPLC coupled with Agilent 6495B triple quadrupole (QqQ) mass spectrometer (Agilent Technologies, Santa Clara, CA, USA). Oligosaccharides were separated through a Waters BEH Amide column (150 mm × 2.1 mm i.d., 2.5 μm particle size) Analytes were detected in negative electrospray ionization mode with dynamic multiple ion monitoring (dMRM). The oligosaccharides were determined by retention time and MRM comparison with the standards. The quantification was conducted by using an external calibration curve.

### 2.7. Analysis of Monosaccharide Composition of Total Polysaccharides (MCTP)

The analysis of monosaccharide composition was adapted from a previous publication [30]. An aliquot of dried pellet and an arabinoxylan polysaccharide standard (QC material) was subjected to acid hydrolysis with 4 M TFA for 1 h at 121 °C in 96-well plates. Afterward, a 10 μL aliquot and a pool of external standards containing 14 monosaccharides were derivatized by adding 100 μL of 0.2 M PMP in methanol, 100 μL of ammonia solution (28–30% *w*/*v*), and heating to 70 °C for 30 min. The derivatized glycosides were dried by vacuum centrifugation and extracted twice with chloroform to remove the excess PMP. A 1 μL aliquot of the aqueous layer was injected into an Agilent 1290 Infinity II UHPLC system equipped with an Agilent Poroshell HPH C18 column (50 mm × 2.1 mm i.d., 1.8 μm particle size) for analysis. Mass spectral analysis was carried out on an Agilent 6495B QqQ mass spectrometer (Agilent Technologies, Santa Clara, CA, USA) operated in positive ion mode while using dMRM. The total monosaccharide content was calculated in each sample by using the external calibration curve.

### 2.8. Analysis of Glycosidic Linkage of Total Polysaccharides (GLTP)

The glycosidic linkage analysis was performed based on a published method [30,31]. In brief, the dried pellet was reconstituted with 1 mL of pure water and incubated in the oven for an hour at 100 °C to solubilize polysaccharides. Under an argon atmosphere, 5 µL aliquots of sample stock solutions were permethylated at room temperature with 5 µL of saturated NaOH, 150 µL of DMSO, and 40 µL of iodomethane. The samples were agitated for 30–50 min between each addition of reagent. The reaction was quenched by 500 µL of ice-cold water. NaOH and DMSO were removed with cold water extractions. Each permethylated sample was extracted by dichloromethane (DCM). The DCM layer contained the permethylated samples, and this layer was collected and dried by vacuum centrifugation. The dried permethylated products were hydrolyzed with 4 M TFA at 100 °C for 2 h, and the released permethylated monosaccharide residues were derivatized with PMP. For analysis, 1 µL of sample was injected and analyzed using an Agilent 1290 Infinity II UHPLC system and an Agilent Zorbax RRHD Eclipse Plus C18 column (150 mm × 2.1 mm i.d., 1.8 μm particle size). To identify glycosidic linkages, the Agilent 6495B QqQ MS was used to compare MRM transitions and retention times with an established library. Fifty-one possible glycosidic linkages were screened, and relative abundances of the detected linkages were calculated.

### 2.9. Principal Component Analysis

The data matrix comprised triplicate samples for six different treatment groups, each containing relative abundances of linkage data from 27 variables (different types of glycosidic linkage). It was exported to MATLAB R2020b (The MathWorks, Natick, MA, USA). In-house MATLAB scripts were utilized to preprocess the data through normalization prior to performing principal component analysis (PCA).

### 2.10. Statistical Analysis

Corn samples from each processing condition were prepared and analyzed in triplicate. All data were reported as the mean ± standard deviation, analyzed using one-way analysis of variance (ANOVA) with significant differences between means determined at *p* < 0.05 and Tukey’s post-hoc test, measured with Origin software, version 2023 (OriginLab Corporation, Northampton, MA, USA).

## 3. Results

### 3.1. Quality Control for AOAC Methods

Results for the NIST SRM and established in-house control were shown in Appendix A. The values of sugars and different fractions of DF in NIST SRM were all within the range of Target ± Uncertainty. Target values (including Mean ± SD and range) of starch in in-house control were obtained after 27 runs on the QC material over 17 years. The levels of starch in this study were within the range of target values.

### 3.2. Carbohydrates in Fresh Sweet Corn

The content of different carbohydrate fractions in fresh bi-color sweet corn, including simple sugars, oligosaccharide, starch and DF, were presented in Table 1. The data were expressed as g or mg per 100 g fresh weight (FW), as these are the units used in USDA FoodData Central (https://fdc.nal.usda.gov/ accessed on 1 September 2023). Three simple sugars (glucose, sucrose and galactose) were identified and quantified because their concentrations were above LOQ. Sucrose is the predominant simple sugar and constitutes over 86% of total simple sugar. Of the oligosaccharides evaluated, only kestose was measurable with a value of 0.15 g/100 g FW. IDF was found to be the major fraction of DF, accounting for 69% of TDF.

### 3.3. Effects of Blanching, Freezing and Canning on Simple Sugars

Blanching and freezing did not alter the sucrose levels but canning led to significant decreases of sucrose in Can 0 and Can 1 samples. Total simple sugars followed the same trend, likely due to sucrose being the major simple sugar (Figure 3A). Compared to the Baseline sample, glucose and fructose decreased in the Blanch and FO 1 samples but remained unchanged in the FC 1 sample compared to the Blanch sample. The levels of the two minor sugars were lower in the two canned samples (Can 0 and Can 1) than Blanch and FO 1 samples (Figure 3B).

### 3.4. Effects of Blanching, Canning and Freezing on Oligosaccharide and Starch

Generally, thermal treatments and freezing reduced kestose levels in sweet corn. While blanching did not significantly alter kestose concentration, freezing and canning resulted in a decrease relative to baseline controls (Figure 4A). Similarly total starch levels decreased in both FC 1 and FO 1 samples when compared to baseline but did not show a significant difference from the Blanched sample. Starch was increased in both Can 0 and Can 1 samples compared to Blanch and Baseline samples (Figure 4B).

### 3.5. Effects of Blanching, Freezing and Canning on Dietary Fiber

Both SDFP and SDFS were found to be the minor components of TDF. Blanching, freezing and canning did not alter the contents of SDFP and SDFS (Figure 5A,B). IDF was the major fraction of TDF, and both IDF and TDF were increased in the Blanch sample from the Baseline sample. Freezing and canning further increased IDF and TDF compared to the Blanch sample (Figure 5C).

### 3.6. Effects of Blanching, Freezing and Canning on Monosaccharide Composition of Total Polysaccharides (MCTP)

Six monosaccharides, glucose, galactose, xylose, arabinose, fructose, and ribose, were found to be the major building blocks of total polysaccharides in the sweet corn sample In the Baseline sample, glucose is the primary monosaccharide of the sweet corn MCTP, counting 85.6% of total monosaccharides, and arabinose, xylose and galactose contributed to 5.3%, 4.8% and 3.7% of total monosaccharides, respectively. Fructose and ribose were only in trace amounts, the two combined only counted for 0.6% of total monosaccharides. The blanching and freezing did not significantly alter glucose and total monosaccharide contents compared to the Baseline sample, but both Can 0 and Can 1 samples had increased glucose and total monosaccharide contents compared to the Baseline sample (Figure 6A). Arabinose and xylose did not differ between the Baseline and five treatment groups, but galactose was higher in Can 0 than in Blanch and FO 1 (Figure 6B).

### 3.7. Effects of Blanching, Freezing and Canning on Glycosidic Linkages of Total Polysaccharides (GLTP)

Of the 51 possible glycosidic linkages being screened, 27 were detected in total polysaccharides of sweet corn. The percentages of relative abundance of the detected GLTP in all sweet corn samples are shown in Figure 7A. Three linkages were found to be the primary ones (>5%), including 4-glucose (60–70%), T-glucose (7–14%) and T-F-arabinose (7–14%). The changes in the relative abundance of these three major linkages under different processing methods are shown in Figure 7B. Food processing profoundly altered the relative abundance of glycosidic linkages. Different linkages underwent different patterns as evidenced by the three major linkages. The principal component analysis (PCA) was conducted to understand the changes in the overall patterns of glycosidic linkages (three replicate analyses of one sweet corn variety and 27 glycosidic linkages) (Figure 8). In this analysis, PC1 (80.34%) and PC2 (18.36%) account for approximately 98.70% of the total variability, showing that the data could be differentiated in this unsupervised model. The baseline group was on the left side, and distinct from other groups. FO1 and the two canned groups are located on the right side, indicating their relatively similar linkage trends. The linkage conditions of the Blanch group were in the middle, suggesting it was a transition group. FC 1 was large due to high large within-group variations (Figure 8). PCA loading plot of the glycosidic linkages of total polysaccharides was also provided as Appendix A.

## 4. Discussion

“Supersweet” varieties of sweet corn were developed from the Shrunken2 (*sh2*) mutation, which increases the sweetness and reduces the conversion of sugar to starch after harvest [32]. These varieties are now commonly found in commercial sweet corn in the US. Therefore, a bi-color Supersweet corn variety was selected for use in this study. The fresh sweet corn contained 6.89 g/100 g FW starch, which is lower than the total simple sugars (10.27 g/100 g FW), confirming the sweet corn used in this study was a *sh2* mutation Supersweet variety [33]. The results of carbohydrate fractions analyzed in the fresh bi-color sweet corn, including simple sugars, starch and DF, are very much in line with previous reports [4,22,34]. As a minor component, oligosaccharides were seldom studied in sweet corn in the past. Kestose was the only oligosaccharide detected in a fresh sweet corn sample. It is a fructooligosaccharide and was reported in sweet corn for the first time. A recent study suggested that kestose supplementation modulated the composition of gut microbiota to improve glucose metabolism in hosts predisposed to obesity [35]. Analysis of DF has been evolving to reflect a wider understanding of the types and physiological functions of DF [36]. The term “dietary fiber” was initially proposed to describe the non-digestible components of plant cells such as cellulose, hemicellulose and lignin [37,38]. RS and non-digestible oligosaccharides have later been included in the definition of DF [36]. In this study, AOAC 2011.25 was selected as it largely complies with the current definition of DF. It measures three fractions of DF, including IDF, SDFP and SDFS, based on the solubility. The total soluble DF (SDF) was calculated by adding SDFP to SDFS. In the literature, DF can also be expressed as low molecular weight DF (LMWSDF) and high molecular weight DF (HMWSDF) based on the molecular size. LMWSDF is basically another term of SDFS that refers to non-digestible oligosaccharide fraction. While HMWSDF was calculated as the combination of IDF and SDFP. Both solubility and molecular size are important characteristics of DF that are related to its physiological functions. As resistant starch was included as part of IDF by this method [39], no separate resistant starch measurement was taken.

When evaluating the effects of food processing, the carbohydrate contents were presented based on the g/100 g dry weight (DW) because moisture levels may impact accuracy in the comparison of processing methods. Effects of blanching, freezing and canning on simple sugars were reported in a few previous publications [22,40,41,42,43], but the data are still very limited, and the results were not consistent and sometimes conflict. Of the three papers examining the effects of blanching, two found that total sugar significantly decreased after blanching [40,41]. The third one found that when comparing the sucrose levels in three cultivars (including two Sugary and one Supersweet cultivars), sucrose content increased from unblanched cobs to 4 min blanched cobs (1.08 to 5.90%, respectively). [42]. Our data showed that blanching in hot water for 3 min did not alter either sucrose or total simple sugar but resulted in decreases in glucose and fructose (Figure 3). The results are largely in agreement with that in another paper published recently by our group, in which similar results were obtained for sucrose, glucose and fructose in sweet corn after boiling (5 min) [22]. The discrepancy between the data from our study and those of others can have different explanations, including the type of sweet corn, processing methods/conditions, and analytical methods. For the three papers mentioned above, different cultivars, including Sugary and Supersweet sweet corn, were used in the experiments. The sweet corn was blanched by different methods with or without kernels on the cob. Notably, each of those papers used a different analytical method. Therefore, direct comparison between these different studies is of limited utility. The effects of freezing and canning on the total sugars were studied in only one paper published by Alan et al. [43]. In this study, seven sweet corn varieties were included, and the frozen and canned samples were processed after the corn was blanched. In general, freezing and canning led to decreases in total sugars, and canned samples contained even lower total sugars than frozen samples. However, the extent of the decrease appeared cultivar-dependent. For certain cultivars, the levels of total sugars were not changed or even higher in frozen sweet corn. Our data are partly in agreement with this paper in terms of the total sugars, but we provided additional information on the individual sugars, which provided more insight into how freezing and canning alter the concentrations of different simple sugars. In our study, when compared to the Blanch sample, freezing with kernels on the cob (FC 1) or off the cob (FO 1) for one month did not alter sucrose concentrations in sweet corn (Figure 3A), but glucose and fructose increased in FC 1 while remained unchanged in FO 1 (Figure 3B). All three simple sugars were significantly reduced after canning (Figure 3), likely due to the degradation during canning and/or the loss in the canning liquid.

Effects of food processing on oligosaccharides in sweet corn have not been previously reported. Our results showed that all food processing methods tended to decrease kestose, though only the two frozen and two canned samples showed statistically significant differences (Figure 4A). In our study, compared to the Baseline sample, the amount of starch did not change after blanching but decreased after freezing and increased after canning (Figure 4B). Many factors impact the effects of thermal processing on starches, such as the location of starch granules, the chemical properties of the starch, the food matrix and the processing methods/conditions. Corn starch granules are in the endosperm of kernels, which may help to explain why starch was stable after blanching. Supersweet corn contained a significantly high ratio of amylose/total starch percentage [44]. Food processing, especially thermal treatment and freezing, may cause the gelatinization and retrogradation processes, which can result in a change in the breakdown of starch thus affecting the measurement of starch. The decreases in starch in the two frozen samples are likely related to the rapid retrogradation of amylose molecules during freezing storage [45], which reduced the enzymatic digestion of the starch. Canning, on the contrary, disrupted starch structure during gelatinization thus leading to greater enzymatic digestion. Except for our study, the effects of freezing and canning on starch in sweet corn were only reported in one other publication [43], in which starch was found to be higher, with no change or lower in frozen samples when compared to the fresh samples, depending on the varieties and harvest years. The findings are quite interesting because they suggested that different cultivars of sweet corn (unfortunately no specific information regarding the gene mutations was provided), or the same cultivar growing at different years, may have different starch profiles and/or possibly food matrix. In this previous study, the levels of starch were almost always lower in canned samples. However, there were several limitations associated with this study. In addition to the lack of information on individual cultivars, only a single value was provided for each sample, and polarimetry was used to determine starch. Polarimetry is an old method that does not reflect the digestion nature of starch and is generally used with fairly pure starch samples. It is worth mentioning that though historically several methods have been used for measuring starch [46], the enzymatic method involving gelatinization of the starch and the digestion with α-amylases and amyloglucosidase has become the method of choice in recent decades [47]. The enzymatic method according to AOAC 996.11 [25] was used in our study to measure starch.

In a previous paper by Makhlouf et al. assessing the impact of processing on DF content in sweet corn, soluble, insoluble, and total fiber did not change significantly after freezing or canning [48]. It was also mentioned in a review article that the canning process, absent some additional physical separation process, did not alter DF in the canned vegetables [49]. However, in our study blanching, and freezing, and canning all resulted in increased IDF and TDF, while SDFP and SDFS were unaffected (Figure 5). Clearly, the increase in TDF was due to the increase in IDF. In sweet corn, IDF largely consists of cell wall constituents (e.g., cellulose and hemicellulose) and resistant starch [50]. As cell wall constituents are generally resistant to food processing, the increases in IDF are probably related to the formation of resistant starch during processing. Many factors can cause the alteration of resistant starch in plant foods after processing, including the types of starch (amylose and amylopectin ratio), food matrix, and processing methods/conditions. Processing or cooking techniques may affect both the gelatinization and retrogradation processes, which can result in an increase or decrease in the resistant starch values from those found in raw foods [51]. In general, resistant starch increases with low-temperature storage [52], which may explain the increase of IDF in FO 1 and FC 1 samples. It is more difficult to generalize the effects of heat moisture treatment on resistant starch. It may cause an increase or a decrease in the resistance of starch depending on the types of food and processing conditions [53]. For instance, repeated autoclaving of wheat starch generated up to 10% resistant starch, and this level of resistant starch was related to the amylose content [52]. In another study, the main mechanism for the formation of resistant starch was identified as the retrogradation of amylose [54]. Supersweet corn has a significantly high amylose/total starch percentage [44]. This could be the reason for increasing IDF content after canning. Interestingly, the data from the current study are different from another recent publication by our group [22], in which both boiling and steaming significantly decreased TDF and resistant starch. One possible reason is the types of sweet corn in the two studies were sourced from different global regions due to growing season limitations. The Supersweet cultivar examined in this study was grown in the United States and sourced directly from the farm, as a commercial producer would. The raw sweet corn for our previous study was purchased from a market in Macau, China to reflect consumer interaction with the food system. American consumers like the crunchy texture of sweet corn, whereas waxy corn has a chewy texture that is preferred by many consumers in Southeast Asia and China [55]. Corns that combine both sweet and waxy flavors are now often preferred and were estimated to make up one-third of the Chinese specialty corn market in 2018 [55,56]. The mutation of Wx gene blocks the biochemical route for amylose production and allows nearly total amylopectin production in maize endosperm. The increasing digestion through gelatinization of amylopectin may be the primary reason for the resistant starch loss after cooking in our pervious study.

Other than simple sugars and oligosaccharides, the major types of polysaccharides (e.g., starch and DF) are commonly measured by non-specific AOAC methods based on the digestibility by α-amylase and amyloglucosidase [36,47]. However, no structural information is provided by these AOAC methods. With regard to the physiological functions of polysaccharides, indigestibility does not equal functionality. The influence of the physiological properties of DF on the gut microbiota is determined by the chemical structures and physical properties, including monosaccharide composition, anomeric configurations, linkage types, linear chain lengths, branch chain compositions, and reducing terminal attachments, food matrix, particle size, viscosity, microstructure, etc. Changes in these properties, even the small changes in the DF structure, may profoundly affect the digestibility, accessibility and utilization of DF by gut microbiota [57,58,59]. Moreover, for starch, whether and how fast it is digested also depends on the structure [60]. To further understand the effects of thermal treatment on the biological activities of polysaccharides, more structural information is needed [61]. In this study, the effects of blanching, freezing and canning on the structures of polysaccharides, including MCTP and GLTP, were explored in sweet corn before and after processing. MCTP analysis revealed that 6 monosaccharides are major building blocks of total polysaccharides, and glucose is the major monosaccharide of MCTP. This is reasonable as glucose is the main building block of polysaccharides in plant foods, including starch and cell wall constituents such as cellulose and hemicellulose. Total monosaccharides increased significantly in the Can 0 and Can 1 samples mainly due to the increase of glucose (Figure 6A). A possible explanation for this phenomenon is that in canned samples, certain water-soluble mass was lost in the canning liquid, while the total polysaccharides remained unchanged in the kernels. Consequently, the glucose and total monosaccharides were proportionally higher on the dry weight basis. The results of processing methods on three minor sugars (arabinose, xylose and galactose) did not show clear trends, which is partly due to the higher variation (Figure 6B).

Analysis of GLTP provided additional structural information regarding changes in the glycosidic linkages during food processing. The glycosidic linkages are key factors in determining the digestion of polysaccharides in the GI tract and the interaction between non-digestible polysaccharides and gut microbiota. The percentages of relative abundance of the 27 detected GLTP were altered profoundly from the Baseline to different processed samples (Figure 7A). The effects were demonstrated by the three major glycosidic linkages (>5%) (Figure 7B). Every glycosidic linkage appeared to display a unique pattern after food processing. One interesting observation is that T-glucose and T-F-arabinose showed almost opposite trends. In addition, PCA was also conducted to understand the changes in the linkage patterns for each processed group. It facilitates visual comparison of notable differences and mitigates subjective decision-making. The analysis shows different groupings of the linkages between Baseline and different processed samples, indicating changes in glycosidic linkages of total polysaccharides among them after food processing.

## 5. Conclusions

This study systematically investigated the impact of blanching, freezing, canning and storage on the carbohydrate composition of sweet corn. The results suggested that different fractions of carbohydrates in sweet corn underwent different changes, affected by the types of carbohydrates and processing methods/conditions. The findings from this study presented new information such as the identification of kestose in sweet corn and provided information to better identify the effects of food processing on carbohydrates in sweet corn. In addition, the impact of thermal treatment methods on the structures of polysaccharides was explored. The findings indicated that monosaccharide composition and glycosidic linkages can be profoundly changed after different processing methods, with or without the changes in total polysaccharides. It is noteworthy that mutations that sweet corn carries can affect the carbohydrate profile. These mutations may impact the results of heat moisture treatment. Further investigation on the changes in monosaccharide composition and glycosidic linkages in each specific fraction of polysaccharides, such as different fractions of DF, is currently ongoing in our lab. Further research may help to explore how the changes of carbohydrates during food processing influence their digestibility and physiological functions in the human body.

## Figures and Tables

**Figure 1 foods-12-03885-f001:**
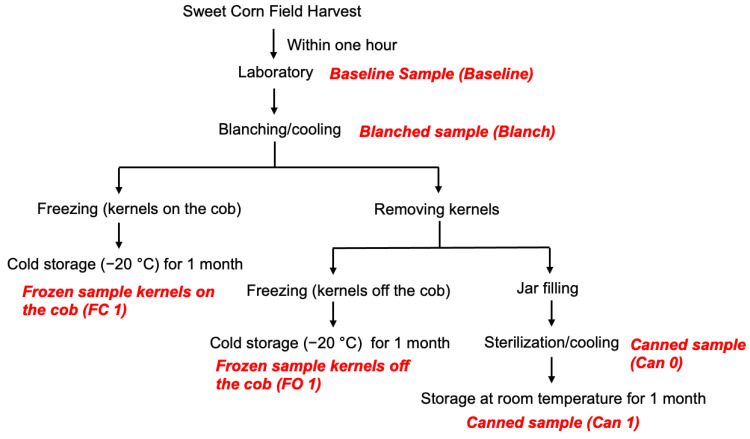
Flow chart of the experimental design.

**Figure 2 foods-12-03885-f002:**
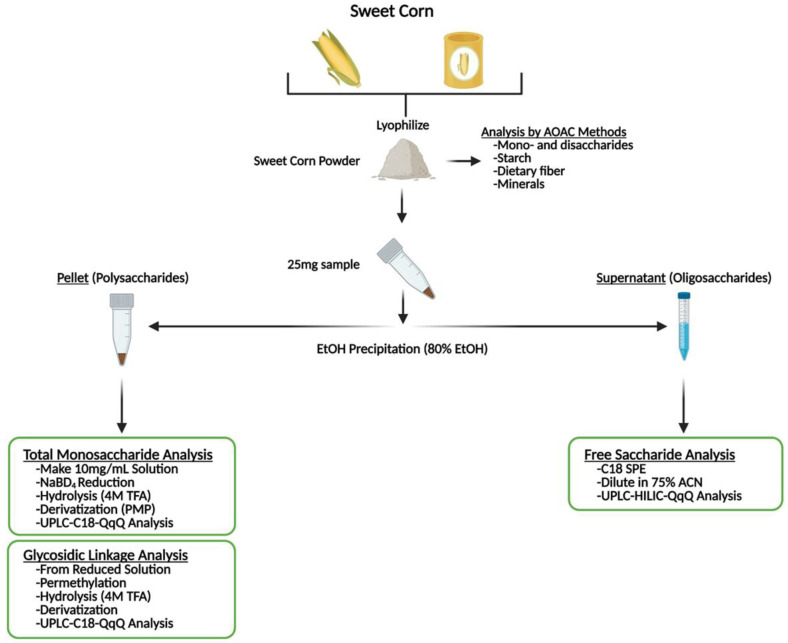
Flow chart of the sample preparation and analyses.

**Figure 3 foods-12-03885-f003:**
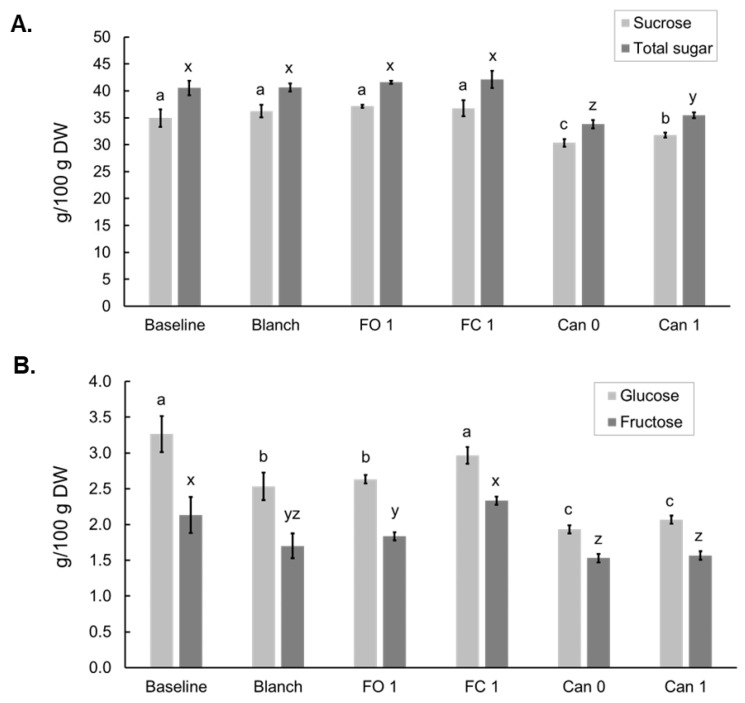
Effects of blanching, freezing and canning on simple sugars in bi-color sweet corn: sucrose and total simple sugars (**A**) and glucose and fructose (**B**). Different letters above the bars indicate statistically significant difference (*p* < 0.05).

**Figure 4 foods-12-03885-f004:**
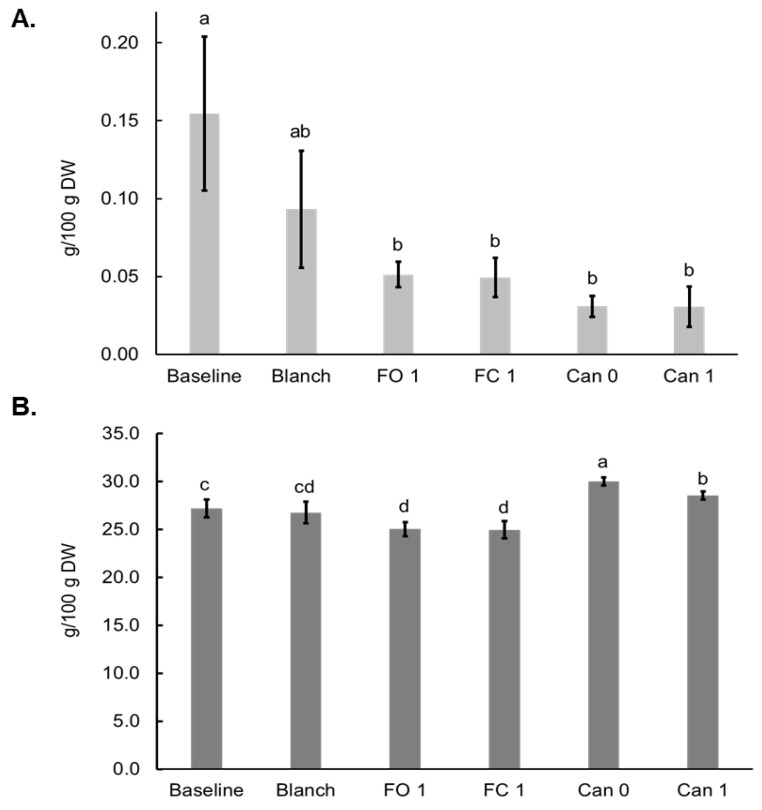
Effects of blanching, freezing and canning on kestose (**A**) and starch (**B**) in bi-color sweet corn. Different letters above the bars indicate statistically significant difference (*p* < 0.05).

**Figure 5 foods-12-03885-f005:**
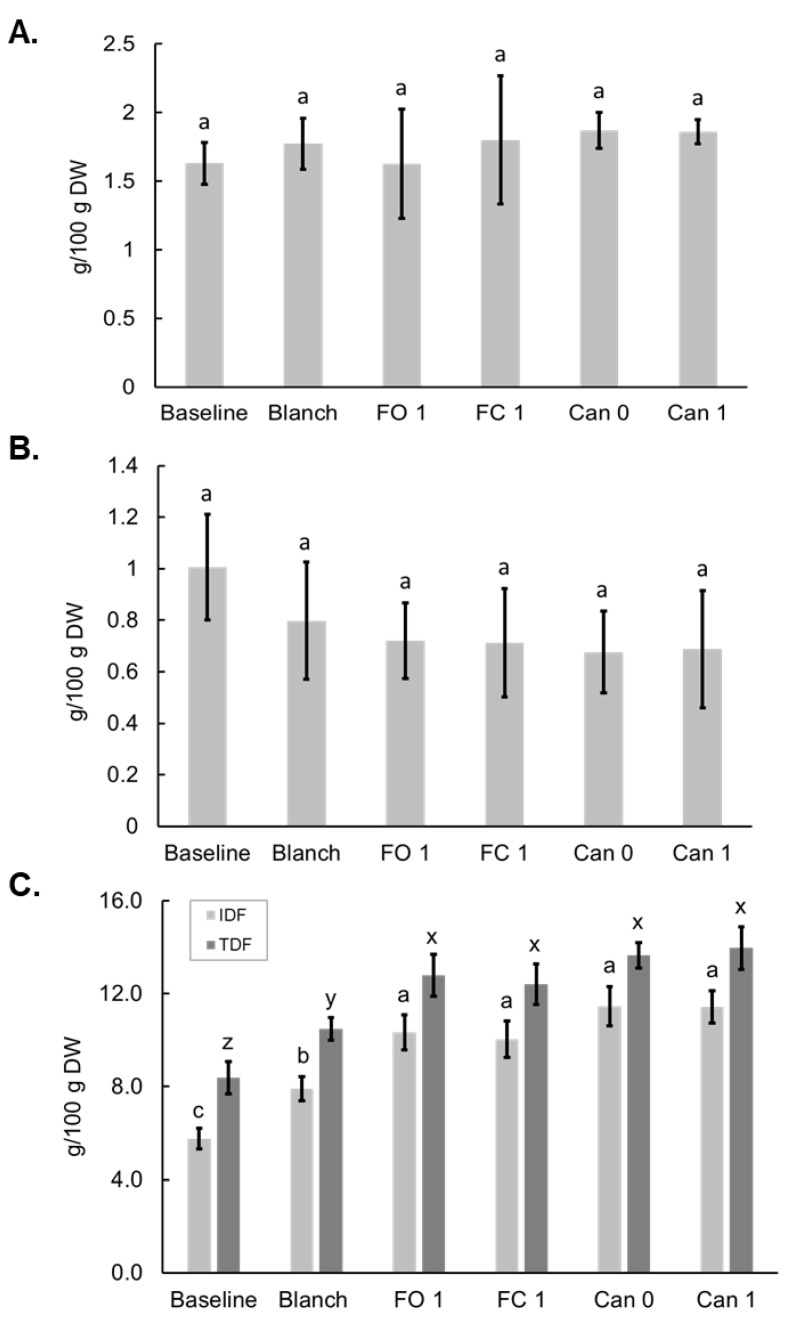
Effects of blanching, freezing and canning on dietary fiber soluble in water but not in 78% alcohol (SDFP) (**A**); dietary fiber soluble in water and 78% alcohol (SDFS) (**B**) and insoluble dietary fiber (IDF) and total dietary fiber (TDF) (**C**) in bi-color sweet corn. Different letters above the bars indicate statistically significant difference (*p* < 0.05).

**Figure 6 foods-12-03885-f006:**
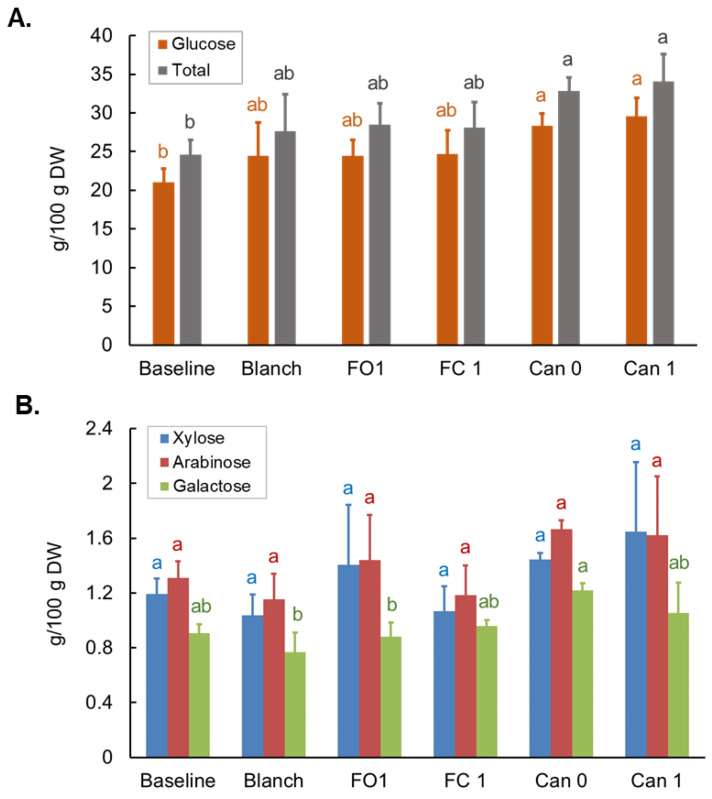
Effects of blanching, freezing and canning on glucose and total monosaccharides (**A**) and xylose, arabinose and galactose (**B**) of total polysaccharides in bi-color sweet corn. Different letters above the bars indicate statistically significant difference (*p* < 0.05).

**Figure 7 foods-12-03885-f007:**
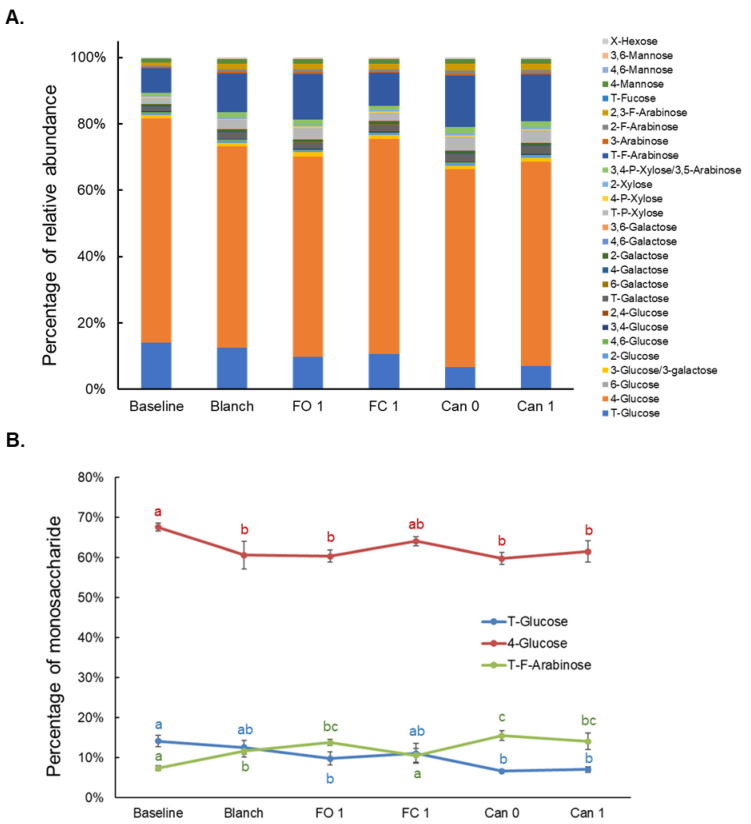
Effects of blanching, freezing and canning on the glycosidic linkages of total polysaccharides (**A**) and on three primary glycosidic linkages of total polysaccharides (**B**). Twenty-seven glycosidic linkages were detected in total polysaccharides of sweet corn. Explanation of the capital letters in linkage notation: T, terminal; P, pyranose; F, furanose; X, unknown position. Different lowercase letters above the lines (**B**) indicate statistically significant differences (*p* < 0.05).

**Figure 8 foods-12-03885-f008:**
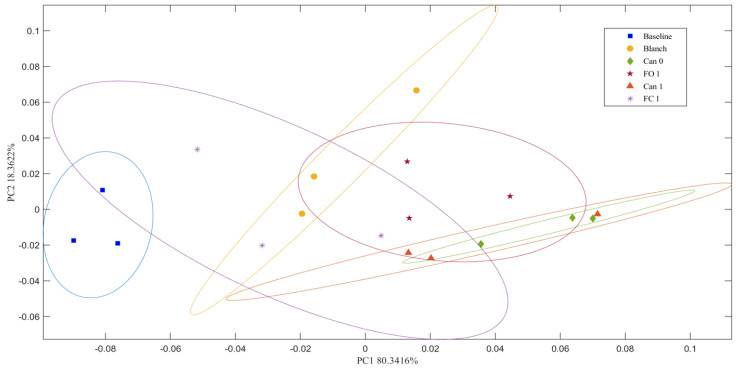
Principal component analysis of glycosidic linkages of total polysaccharides.

**Table 1 foods-12-03885-t001:** Carbohydrates in fresh bi-color sweet corn.

Carbohydrate Fraction	Concentration (g/100 g, Fresh Weight)
Glucose	0.83 ± 0.06
Sucrose	8.85 ± 0.42
Fructose	0.52 ± 0.08
Total sugars	10.27 ± 0.34
Starch	6.89 ± 0.23
Kestose	0.15 ± 0.05
Insoluble dietary fiber (IDF)	1.46 ± 0.11
Soluble dietary fiber precipitate (SDFP)	0.41 ± 0.04
Dietary fiber soluble in water and alcohol (SDFS, LMWDF)	0.25 ± 0.05
Total soluble fiber (SDFP + SDFS)	0.67 ± 0.09
HMWDF (IDF + SDFP)	1.87 ± 0.13
Total dietary fiber (IDF + SDFP + SDFS)	2.13 ± 0.18

## Data Availability

The data used to support the findings of this study can be made available by the corresponding author upon request.

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
