# Peer review of "Effects of Blanching, Freezing and Canning on the Carbohydrates in Sweet Corn"

_foods, 2023, doi:10.3390/foods12213885_

Round 1

Reviewer 1 Report

“Effects of Blanching, Freezing and Canning on the Carbohydrates in Sweet Corn” is a manuscript submitted to Foods, holding a manuscript number 2659418. It is a study on the changes in carbohydrates during various processing of sweet corn; the authors did multiple efforts to see the effects on different carbohydrates in sweet corn during blanching, freezing and canning. The results of this study may have commercial values in terms of sweet corn market as well as in diabetes arena. The authors did well-explained discussion on their obtained results. The authors also concluded that different fractions of carbohydrates in sweet corn undergo different changes, affected by the types of carbohydrates and processing methods and / or processing conditions. However, I found following issues as the shortcomings for this manuscript:

Abstract: The abstract is too long; please revise it to concise it to 200 words or 250 words (maximum)?

Section 2.5: “In brief, a 25 mg aliquot of lyophilized sweet corn was weighed into a 1.5 mL screw cap Eppendorf tube and then reconstituted with 80% ethanol.” Please explain this in detail.

Section 2.6: “Samples were prepared by creating a 10 mg/mL stock solution and diluting it with water, then diluting further into 75% acetonitrile.” Please explain this in detail.

Fig. 5C: The TFD and IFD were found to increase with these processing (blanching, freezing and canning); can you please explain the reason behind such results?

None.

Author Response

Abstract: The abstract is too long; please revise it to concise it to 200 words or 250 words (maximum)?

- The abstract was revised to 250 words.

Section 2.5: “In brief, a 25 mg aliquot of lyophilized sweet corn was weighed into a 1.5 mL screw cap Eppendorf tube and then reconstituted with 80% ethanol.” Please explain this in detail.

- The description of the method was revised.

Section 2.6: “Samples were prepared by creating a 10 mg/mL stock solution and diluting it with water, then diluting further into 75% acetonitrile.” Please explain this in detail.

- The description of the method was revised.

Fig. 5C: The TFD and IFD were found to increase with these processing (blanching, freezing and canning); can you please explain the reason behind such results?

– This is a very interesting question. In our manuscript, in the 4th paragraph of “Discussion”, we proposed that the increase of TDF was largely caused by the increase of IDF, and the increase of IDF was likely due to the increase of resistant starch.

Reviewer 2 Report

Figures - consider changing colors on the plots (fig. 3,4,5,6b) error bars are invisible on black. Explain the meaning of the letters (also check their sequence - e.g. in the case of fig. 3a the letter z seems to be between x and y, and the letter c between a and b, which does not seem to be a standard way of labelling, fig 4b has similar problems),use spaces in the units on the axes (100 g instead of 100g).

Reconsider connecting fig 6b with 6a (the data are in fact repeated), alternatively you can report the percentage of individual sugars as on fig 7b, which would allow to compare the levels of sugars determined by MCTP and GLTP methods.

Explain the abbreviations used on fig 7a (e.g. T-glucose stands for terminal glucose group and not total glucose, which is sometimes used elsewhere). I would find it much easier to follow if different types of sugar residues are only differing in their shade but not major color (e.g. decreasing intensity of blue for glucose, and green for galactose).

The results of PCA would be easier to interpret if you add the information about the eigenvectors (it could be appended at the end of the article as a table or a loading plot).

I would not say that polarymetric method of starch measurement is inferior to enzymatic one in general. However as the aim of your study is to measure the constituents important for human physiology the choice is quite obvious (on the other hand if you would like to know the percentage of starch for its industrial isolation, similar choice would not be the best).

The loss of dry matter during canning explains the apparent increase in starch or insoluble fiber, but requires a little more attention - is canned corn supposed to be eaten with or without the soaking liquid? If so, the liquid composition should also be checked.

Some typos are marked in the attached pdf.

Author Response

Figures - consider changing colors on the plots (fig. 3,4,5,6b) error bars are invisible on black. Explain the meaning of the letters (also check their sequence - e.g. in the case of fig. 3a the letter z seems to be between x and y, and the letter c between a and b, which does not seem to be a standard way of labelling, fig 4b has similar problems),use spaces in the units on the axes (100 g instead of 100g).

- We agree with the reviewer on suggestions. The colors on the plots were changed. The explanations of the meaning of the letters were added in figure legends of Figure 3,4,5,6 and 7. The letters were rearranged in a more logical way, with a to c and x to z for values increased. Spaces in the units on the axes were added.

Reconsider connecting fig 6b with 6a (the data are in fact repeated), alternatively you can report the percentage of individual sugars as on fig 7b, which would allow to compare the levels of sugars determined by MCTP and GLTP methods.

- We agree with the reviewer that the data is repeated. We deleted 6a, and in the revised manuscript, we presented the data of glucose and total monosaccharides in 6a, and three minor sugars galactose, xylose, arabinose in 6b. The concentrations of the other two sugars fructose and ribose were extremely low (combined only contribute to 0.6% of total monosaccharides). They were not presented in the figures. An explanation was added to the text (page 9).

Explain the abbreviations used on fig 7a (e.g. T-glucose stands for terminal glucose group and not total glucose, which is sometimes used elsewhere). I would find it much easier to follow if different types of sugar residues are only differing in their shade but not major color (e.g. decreasing intensity of blue for glucose, and green for galactose).

- The explanation of the letter of linkage notation were added in figure legend of Figure 7. It is a great idea to try to present the data in different shades and we tried a few ways. However, as most of the linkages (19 out of 27) contribute less than 1% of total linkage, the figures looked pretty empty. Except for a few major ones, it is difficult to see the differences between different treatment groups. Therefore, we feel that the current bar graph, though not perfect, is a better choice.

The results of PCA would be easier to interpret if you add the information about the eigenvectors (it could be appended at the end of the article as a table or a loading plot).

- The eigenvectors were added as Supplemental Figure 1, and it was also mentioned in the text (page 11).

I would not say that polarymetric method of starch measurement is inferior to enzymatic one in general. However as the aim of your study is to measure the constituents important for human physiology the choice is quite obvious (on the other hand if you would like to know the percentage of starch for its industrial isolation, similar choice would not be the best).

- This is a great insight. We completely agree with the reviewer.

The loss of dry matter during canning explains the apparent increase in starch or insoluble fiber, but requires a little more attention - is canned corn supposed to be eaten with or without the soaking liquid? If so, the liquid composition should also be checked.

- Yes, the reviewer brought up an excellent point. Though some consumers may eat the soaking liquid, normally most consumers discard the liquid. Another reason to discard the liquid is that in industrial processing, a significant amount of sodium is added. It is considered as a sodium reduction suggestion to not eat the liquid. Therefore, to reflect the real-world scenario, we did not measure the liquid.

Some typos are marked in the attached pdf.

- We greatly appreciate the time that the reviewer took to correct the typos. All corrections were accepted.